

# A framework for studying a quantum critical metal in the limit $N_f \to 0$

Petter Säterskog[1,2*]

**1** Nordita, KTH Royal Institute of Technology and Stockholm University,
Roslagstullsbacken 23, SE-106 91 Stockholm, Sweden
**2** Institute Lorentz $\Delta$ITP, Leiden University,
PO Box 9506, Leiden 2300 RA, The Netherlands

★ petter.saterskog@su.se

## Abstract

We study a model in 1+2 dimensions composed of a spherical Fermi surface of $N_f$ flavors of fermions coupled to a massless scalar. We present a framework to non-perturbatively calculate general fermion $n$-point functions of this theory in the limit $N_f \to 0$ followed by $k_F \to \infty$ where $k_F$ sets both the size and curvature of the Fermi surface. Using this framework we calculate the zero-temperature fermion density-density correlation function in real space and find an exponential decay of Friedel oscillations.

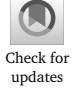

# 1 Introduction

Quantum critical metals are systems of electrons at finite density near a zero-temperature phase transition. The low-energy degrees of freedom of such systems contain—in addition to the electronic quasiparticles—fluctuations of the critical order parameter.

One example of such a quantum critical point (QCP) is the Ising-nematic transition. This transition has been seen in the vicinity of the, yet to be understood, strange metallic and high-$T_c$ superconducting phases of cuprates and iron-pnictides [1–4].

The essential ingredients of an effective field theory description of quantum critical metals are a fermionic field representing the electronic quasiparticles and a massless bosonic field representing the critical order parameter fluctuations. The bosonic field has low-energy fluctuations either at zero momentum, or at a finite momentum $Q$, depending on if the order parameter expectation value in the ordered phase has zero momentum (Ising-nematic, ferromagnetic) or a finite momentum (spin/charge density waves, antiferromagnetic).

Although well understood in three spatial dimensions [5, 6], quantum critical metals in two dimensions, which is the relevant dimensionality for the cuprates and iron pnictides, have eluded a full theoretical understanding. This is due to the interaction between the fermionic and bosonic fields being relevant in the IR and thus preventing the use of perturbation theory. Additionally, the finite density of fermions generally gives rise to the fermion sign problem when using Monte-Carlo methods [7].

Several approximations have been employed to find cases where some of the physics can be understood. Many approaches extend the theory to get a new expansion parameter, $\epsilon$, such that the system can be treated for small values of $\epsilon$. The considered $\epsilon$ are typically not small in the physical systems we are ultimately interested in so these approaches can only give limited insights at the moment.

One example of this approach is to not study the model in 2 dimensions, but in $3-\epsilon$ dimensions [8, 9]. Since the theory can be studied perturbatively in 3 dimensions we can then use $\epsilon$ as a small expansion parameter. Other approaches extend the field content of the models. Quantum critical metals in the limit of many fermion flavors have been studied extensively for both $Q = 0$ [10–12] and $Q \neq 0$ [13]. Here the small parameter is given by the inverse of the number of fermionic flavors, $\epsilon = 1/N_f$. Another approach is to study a matrix large-$N$ limit, see e.g. [14–17]. Here the boson is a matrix and the fermion a vector, both transforming under a global $SU(N)$ flavor symmetry under which the boson transforms in the adjoint representation and the fermion in the fundamental. Here the expansion parameter is $\epsilon = 1/N$. The matrix large-$N$ limit suppresses all but the planar diagrams. This suppresses all quantum corrections from the fermion onto the boson, yet the fermion receives non-perturbative quantum corrections from the boson. These corrections are limited in that only planar diagrams contribute.

Another approach is the vector small-$N_f$ limit. Similar to the matrix large-$N$ limit this removes all back-reaction from the fermion onto the boson. In contrast to the matrix large-$N$ limit, this limit keeps all crossed diagrams giving corrections to the fermion. Only diagrams containing fermionic loops are suppressed. The diagrams that survive the matrix large-$N$ limit are therefore a strict subset of those of the $N_f \to 0$ limit. The small $N_f$ limit has been studied in different forms. It is natural to study quantum critical metals at energies much smaller than the Fermi momentum scale set by the chemical potential. However, neither the vector small-$N_f$ limit nor the matrix large-$N$ limit commutes with this low-energy limit. This means that important corrections from the fermions onto the boson, so-called Landau-damping corrections, can not be seen as we take $\epsilon \to 0$ first. Similarly, but more subtly, issues related to this low-energy limit occur in the vector large-$N_f$ limit [10]. Early works in the small-$N_f$ limit used an already explicitly Landau-damped boson and calculated the real space fermion two-point

function [18–21]. This takes into account some of the higher-order in $N_f$ corrections, but not in a systematic way. A more recent study considers the momentum space fermion two-point function in the strict small-$N_f$ limit without any Landau-damping corrections [22]. Surprisingly, the interaction changes the fermion dispersion to become non-monotonic and part of the Fermi surface splits off. A follow up to this work was subsequently done [23] where—as in the earlier works—some of the Landau-damping effects were incorporated, but now in momentum space and systematically by considering a particular simultaneous limit, $k_F \to \infty$, $N_f \to 0$, $N_f k_F$ constant.

In this paper we expand upon the work in [22] to better understand the physics of the peculiar state found in the strict $N_f \to 0$ limit of a $Q = 0$ quantum critical metal. We do this by developing a framework to analytically calculate higher-point fermion correlation functions in real space. By doing this we get more probes of the system and it allows us to use this to search for possible instabilities in the future. Here, we further use this framework to calculate the non-perturbative fermion density-density correlation function of the $N_f \to 0$ quantum critical metal, which is the major new result of this paper. The correlator shows oscillations at wavevector $2k_F$, twice the Fermi momentum, due to the presence of a Fermi surface. These oscillations are contrary to Friedel oscillations found in Luttinger and Fermi liquids not decaying with a power-law at zero temperature, but they are exponentially decaying with the decay length set by the fermion-boson interaction strength.

The paper is organized as follows. In Section 2 we present the framework for calculating general $n$-point functions. In Section 3 we apply this framework to calculate the real space fermion two-point function and the fermion density-density correlator. We discuss the different processes that contribute to these result and compare to some earlier works on related models. We give some conclusions in Section 4 and in the appendix we expand our results in the coupling constant and verify agreement with perturbation theory up to two loops.

## 2   Setup and calculation of fermion $n$-point functions in the $N_f \to 0$ limit

As in the previous works mentioned in the introduction, we study a toy model containing only the necessary ingredients to capture the qualitative behavior of the strongly coupled quantum critical metal. We limit ourselves to the $Q = 0$ case and additionally impose zero boson self-interaction in the bare action. The $N_f \to$ limit then stops such a term from developing. We consider spin-less fermions and impose rotational and translational symmetry. We consider the following action:

$$S = \int d\tau d^2 x \left[ \psi_i^\dagger \left( \partial_\tau - \frac{\nabla^2}{2m} - \mu \right) \psi^i + \frac{1}{2} (\partial_\tau \phi)^2 + \frac{1}{2} (\nabla \phi)^2 - \lambda \phi \psi_i^\dagger \psi^i \right]. \tag{1}$$

The index $i$ takes values $1...N_f$. Coordinates have been chosen such that the boson velocity is one, $c = 1$. In [22] the authors find that for the case of equal Fermi velocity $v_F$ and boson velocity we get a considerably simpler result. The limit of $v_F \to c$ is found to be continuous and qualitatively similar to the case of $0 < v_F < c$ ($c < v_F$ has not been studied yet for $N_f \to 0$) for the two-point function. The cases of $v_F/c \to 0$ or $\infty$ are qualitatively different, however, we believe that $v_F/c = 1$ captures the physics of $v_F \sim c$ also for general $n$-point functions. We therefore exclusively consider this case since it makes analytical calculations much simpler.[1] The techniques that we present here for calculating expectation values in the $N_f \to 0$ limit do

---

[1]The point $v_F = c$ actually results in a type of Fermi surface patch-Lorentz invariance that can be used to bootstrap many properties of both the $N_f \to 0$ and the matrix $N \to \infty$ theories. This is due to be submitted by the author of this paper [24].

not crucially depend on these velocities being the same and much of the calculation can be done for a general $v_F \sim c$. It is only a final set of integrals that for the general case $v_F \neq c$ would likely only be numerically amenable whereas for $v_F = c$, we can find closed-form expressions.

We start by adding sources for the fermion to the action in (1), $\int d^3z(J^i\psi_i^\dagger + J_i^\dagger\psi^i)$, and perform the fermionic path integral to get the generating functional:

$$Z[J^\dagger, J] = \int D\phi \exp\left(-S_{\text{det}}[\phi] - S_b[\phi] - \int d^3z d^3z' J_i^\dagger(z)G_j^i[\phi](z,z')J^j(z')\right) \quad (2)$$

where

$$S_{\text{det}}[\phi] = -\text{tr}\log G_j^i[\phi](z,z') = -N_f\,\text{tr}\log G[\phi](z,z') \quad (3)$$

and $G_j^i[\phi](z,z')$ is the fermion Green's function with a background field $\phi$. We use a simple $z$ to denote $(\tau, x, y)$. This determinant action vanishes for $N_f \to 0$ and this term can thus be neglected since we work to leading order in small $N_f$. The determinant is responsible for all fermionic loops in a perturbative expansion of this theory. By differentiating with respect to the sources and setting them to vanish we find

$$\lim_{N_f \to 0}\langle\psi_{i_1}^\dagger(z_1)\psi^{j_1}(w_1)...\psi_{i_n}^\dagger(z_n)\psi^{j_n}(w_n)\rangle =$$

$$Z[0]^{-1}\int D\phi \sum_{\sigma \in S_n}\prod_k \text{sgn}(\sigma)G_{j_{\sigma_k}}^{i_k}[\phi](z_k, w_{\sigma_k})e^{-S_b}. \quad (4)$$

Here the sum is over permutations of the integers $1, 2, ..., n$ and $\text{sgn}(\sigma)$ is the parity of the permutation $\sigma$.

## 2.1 Background-field fermion two-point function

We will here calculate the background field fermion Greens function for a general background field $\phi$. We do this while keeping in mind that we will later perform the above integral over $\phi$ and that we are only interested in energies small compared to $k_F$. The background field Greens function is defined through

$$\left(-\partial_{\tau_1} + \frac{\nabla_1^2}{2k_F} + \frac{k_F}{2} + \lambda\phi(z_1)\right)G_j^i[\phi](z_1, z_2) = \delta^3(z_1 - z_2)\delta_{ij}. \quad (5)$$

In momentum space $k = (\omega, k_x, k_y)$,

$$G(z_1, z_2) = \int \frac{d^3k_1 d^3k_2}{(2\pi)^6} e^{i(-\omega_1\tau_1 + k_{x1}x + k_{y1}y_1) - i(-\omega_2\tau_2 + k_{x2}x_2 + k_{y2}y_2)}G(k_1, k_2), \quad (6)$$

we have

$$\left(i\omega_1 - \frac{k_1^2}{2k_F} + \frac{k_F}{2}\right)G[\phi](k_1, k_2) + \lambda\int \frac{d^3k'}{(2\pi)^3}\phi(k')G[\phi](k_1 - k', k_2) = (2\pi)^3\delta(k_1 - k_2). \quad (7)$$

For momenta $k_2$ in the vicinity of a point $\hat{n}k_F$ on the Fermi surface we can approximate this as

$$(i\omega_1 - \hat{n}\cdot k_1 + k_F)G_{\hat{n}}[\phi](k_1, k_2) + \lambda\int \frac{d^3k'}{(2\pi)^3}\phi(k')G_{\hat{n}}[\phi](k_1 - k', k_2) = (2\pi)^3\delta^3(k_1 - k_2). \quad (8)$$

Fourier transforming back to real space this now reads

$$\left(-\partial_{\tau_1} + i\hat{n}\cdot\nabla_1 + k_F + \lambda\phi(z_1)\right)G_{\hat{n}}[\phi](z_1, z_2) = \delta^3(z_1 - z_2). \quad (9)$$

The solution to this first order PDE can be written

$$G_{\hat{n}}[\phi](z_1, z_2) = f_{\hat{n}}(z_1 - z_2) \exp\left(ik_F\hat{n}\cdot(z_1 - z_2) + \lambda\int d^3z\phi(z)(f_{\hat{n}}(z - z_1) - f_{\hat{n}}(z - z_2))\right) \quad (10)$$

where $f_{\hat{n}}(z) = \delta(\hat{m}\cdot z)(2\pi)^{-1}/(i\hat{n}\cdot z - \tau)$. $\hat{m}$ is a spatial unit vector perpendicular to $\hat{n}$.

This solution is now only valid for momenta close to $\hat{n}k_F$ but since it is written in real space this statement might seem confusing. $G[\phi]$ should be viewed as an operator on fields that obeys the operator equation (5). $G_{\hat{n}}[\phi]$ is an approximation to this operator that is valid when acting on fields with momentum components close to $k_F\hat{n}$. The operator $G_{\hat{n}}[\phi]$ can be represented in either real or momentum space.

So far this calculation has paralleled that of [22] albeit in the true real space coordinates instead of the coordinates conjugate to patch momentum coordinates used there. $G_{\hat{n}}[\phi]$ is all we need to calculate the fermion two-point function. However, to calculate general $n$-point functions we cannot restrict ourselves to having all fermion momenta in a single patch of the Fermi surface, i.e. in the vicinity of a single point. For low energies and long wavelengths we can still restrict ourselves to momenta close to the Fermi surface, but we must include all directions.

We now construct an operator $G_{\mathrm{IR}}[\phi]$ that approximates $G[\phi]$ well everywhere close to the Fermi surface. See Fig. 1 for a comparison between the approximations $G_{\hat{n}}[\phi]$ and $G_{\mathrm{IR}}[\phi]$. We do this by projecting out momentum components in different directions and applying the corresponding $G_{\hat{n}}[\phi]$. We use the resolution of identity,

$$\delta^3(z, z') = \delta(\tau, \tau')\int\frac{d^2k}{(2\pi)^2}e^{i\left(k_x(x-x')+k_y(y-y')\right)}$$
$$= \delta(\tau, \tau')\int\frac{dk\,k\,d\theta}{(2\pi)^2}e^{ik\hat{n}(\theta)\cdot(z-z')}. \quad (11)$$

Operating with $G_{\mathrm{IR}}[\phi]$ on identity we have

$$G_{\mathrm{IR}}[\phi](z_1, z_2) = \int d^3z'\int\frac{dk\,k\,d\theta}{(2\pi)^2}G_{\mathrm{IR}}(z_1, z')[\phi]\delta(\tau', \tau_2)e^{ik\hat{n}(\theta)\cdot(z'-z_2)}$$
$$= \int d^3z'\int\frac{dk\,k\,d\theta}{(2\pi)^2}G_{\hat{n}(\theta)}[\phi](z_1, z')\delta(\tau', \tau_2)e^{ik\hat{n}(\theta)\cdot(z'-z_2)}$$
$$= \int d^3z'\int\frac{dk\,k\,d\theta}{(2\pi)^2}f_{\hat{n}(\theta)}(z_1 - z')\delta(\tau', \tau_2)$$
$$\times e^{i\hat{n}(\theta)\cdot\left(k_F(z_1-z')+k(z'-z_2)\right)}e^{\lambda I_{\hat{n}(\theta)}[\phi](z_1, z')} \quad (12)$$

where we have used that the action of $G_{\mathrm{IR}}[\phi]$ and $G_{\hat{n}}[\phi]$ are the same when acting on a momentum mode close to $k_F\hat{n}$. Here we have defined

$$I_{\hat{n}(\theta)}[\phi](z_1, z_2) = \int d^3z\phi(z)(f_{\hat{n}}(z - z_1) - f_{\hat{n}}(z - z_2)). \quad (13)$$

In the large $k_F$ limit we can perform these integrals using a saddle point approximation. Here we make use of the fact that $\phi$ will not contain frequencies of order $k_F$ for configurations relevant in the large $k_F$ limit. In principle we could first integrate out $\phi$ and then perform the saddle point approximation. However, the result turns out to be the same and the calculation is more instructive done in the other order. We make the change of variable $z' = z_1 + (\eta\hat{n}(\theta) + v\hat{m}(\theta), \sigma)$

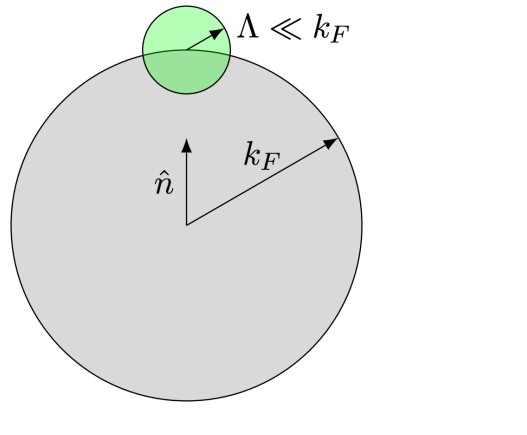

(a) $G_{\hat{n}}$ region of validity

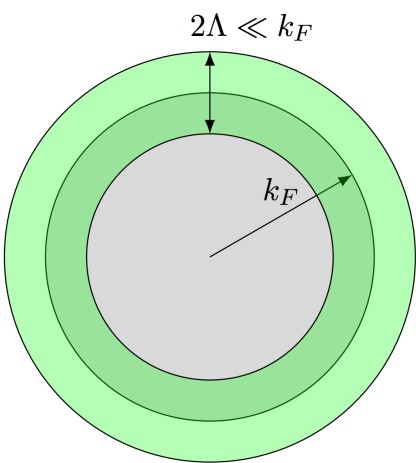

(b) $G_{\mathrm{IR}}$ region of validity

Figure 1: The green areas indicate the regions in momentum space where the approximate background-field Green's functions $G_{\hat{n}}[\phi]$ (a) and $G_{\mathrm{IR}}[\phi]$ (b) are accurate. The boundary of the gray area is the Fermi surface.

$$G_{\mathrm{IR}}[\phi](z_1, z_2) = \int d\eta\, d\nu \int \frac{dk\, k\, d\theta}{(2\pi)^3} \frac{\delta(\nu)}{-i\eta + \tau_2 - \tau_1} e^{i((k - k_F)\eta + k\hat{n}(\theta)\cdot(z_1 - z_2))} \times$$
$$\times\, e^{\lambda I_{\hat{n}(\theta)}[\phi](\tau_1, z_1; \tau_2, z_1 + \eta\hat{n}(\theta))}. \tag{14}$$

The first exponential oscillates faster in $\eta$ than any other factor of the integrand unless $|k - k_F| \ll k_F$. The dominant contribution to the integral will therefore be from $k \approx k_F$. In this $k$-region, the first exponential oscillates rapidly in $\theta$ (since $k_F |z_2 - z_1| \gg 1$), except for the two points where $\hat{n}(\theta)$ is parallel or anti-parallel to $z_{12} = z_2 - z_1$. We therefore perform saddle-point expansions around these two points and perform the $\theta$ integral to obtain

$$G_{\mathrm{IR}}[\phi](z_1, z_2) = \int d\eta \int \frac{dk}{(2\pi)^{5/2}} \frac{1}{-i\eta + \tau_2 - \tau_1} e^{i(k - k_F)\eta} \sqrt{k/|z_{12}|}$$
$$\times \left( e^{-ik|z_{12}| + i\pi/4 + \lambda I_{\hat{z}_{12}}[\phi](\tau_1, z_1; \tau_2, z_1 + \eta\hat{z}_{12})} + e^{ik|z_{12}| - i\pi/4 + \lambda I_{-\hat{z}_{12}}[\phi](\tau_1, z_1; \tau_2, z_1 - \eta\hat{z}_{12})} \right). \tag{15}$$

Next we proceed with the $k$ integral. It is of the form $\int_0^\infty dk\, e^{ikz} \sqrt{k}$ and diverges. In principle these integrals should be performed last for convergence but we can introduce a small imaginary component to $\eta$ so that the Fourier transform integral is regularized. In the end the result is independent of the imaginary part so we can remove it. This amounts to using the result

$$\int_0^\infty dk\, e^{ikz} \sqrt{k} \to \frac{\sqrt{\pi}}{2(-iz)^{3/2}} \tag{16}$$

for these integrals. We then have

$$G_{\mathrm{IR}}[\phi](z_1, z_2) = \int \frac{d\eta}{(2\pi)^{5/2}} \frac{1}{-i\eta + \tau_2 - \tau_1} e^{-ik_F \eta} \frac{\sqrt{\pi}}{2\sqrt{|z_{12}|}}$$
$$\times \left( \frac{e^{i\pi/4 + \lambda I_{\hat{z}_{12}}[\phi](\tau_1, z_1; \tau_2, z_1 + \eta\hat{z}_{12})}}{(-i(\eta - |z_{12}|))^{3/2}} + \frac{e^{-i\pi/4 + \lambda I_{-\hat{z}_{12}}[\phi](\tau_1, z_1; \tau_2, z_1 - \eta\hat{z}_{12})}}{(-i(\eta + |z_{12}|))^{3/2}} \right). \tag{17}$$

Next we integrate $\eta$ and take the large $k_F$ limit. We see that this is the high frequency limit of the Fourier transform in $\eta$. For $|\tau_2 - \tau_1| \gg 1/k_F$, the high frequency part of this function is dominated by the singularities at $\eta = \pm |z_{12}|$. We can expand around them to get the leading large $k_F$ limit:

$$
\begin{aligned}
G_{\text{IR}}[\phi](z_1, z_2) = & \frac{\sqrt{k_F}}{(2\pi)^{3/2}\sqrt{|z_{12}|}} \\
& \times \left( \frac{e^{-ik_F|z_{12}|+i\pi/4+\lambda I_{\hat{z}_{12}}[\phi](\tau_1,z_1;\tau_2,z_2)}}{-i|z_{12}| + \tau_2 - \tau_1} + \frac{e^{ik_F|z_{12}|-i\pi/4+\lambda I_{-\hat{z}_{12}}[\phi](\tau_1,z_1;\tau_2,z_2)}}{i|z_{12}| + \tau_2 - \tau_1} \right) \\
= & \frac{2\sqrt{k_F}}{(2\pi)^{3/2}\sqrt{|z_{12}|}} \text{Re}\left( \frac{e^{-ik_F|z_{12}|+i\pi/4+\lambda I_{\hat{z}_{12}}[\phi](\tau_1,z_1;\tau_2,z_2)}}{-i|z_{12}| + \tau_2 - \tau_1} \right).
\end{aligned}
\tag{18}
$$

These two terms can be understood as momentum modes of momenta parallel and anti-parallel to $z_2 - z_1$ giving the dominant contributions to $G_{\text{IR}}[\phi](z_1, z_2)$. These two contributions couple in different ways to the background field $\phi$.

## 2.2 Integrating over $\phi(z)$

We now have an expression for the background field fermion two-point function that we can substitute into Eq. (4). The next step is to integrate over the field $\phi$. Eq. (4) gives a product of $G_{\text{IR}}[\phi]$ so in performing the $\phi$ integral we will need to evaluate expressions like

$$
H_\lambda(\{\hat{n}_i\}, \{z_i\}, \{w_i\}) = Z[0]^{-1} \int D\phi \exp\left( \lambda \sum_i I_{\hat{n}_i}[\phi](z_i, w_i) - S_b[\phi] \right)
\tag{19}
$$

where $\hat{n}_i$ is either parallel or antiparallel to the spatial part of $w_i - z_i$. The result of this Gaussian path-integral is

$$
\begin{aligned}
H_\lambda = & \exp\left( \frac{\lambda^2}{2} \int d^3Z d^3W \left( \sum_i f_{\hat{n}_i}(z_i - Z) - f_{\hat{n}_i}(w_i - Z) \right) \right. \\
& \left. \times \left( \sum_j f_{\hat{n}_j}(z_j - W) - f_{\hat{n}_j}(w_j - W) \right) G_b(Z - W) \right) \\
= & \exp\left( \lambda^2 \sum_{i<j} \left( h_{\hat{n}_i,\hat{n}_j}(z_j - z_i) - h_{\hat{n}_i,\hat{n}_j}(z_j - w_i) + \right. \right. \\
& \left. \left. -h_{\hat{n}_i,\hat{n}_j}(w_j - z_i) + h_{\hat{n}_i,\hat{n}_j}(w_j - w_i) \right) - \lambda^2 \sum_i h_{\hat{n}_i,\hat{n}_i}(z_i - w_i) \right)
\end{aligned}
\tag{20}
$$

where $h$ is defined as

$$
h_{\hat{n}_1,\hat{n}_2}(z) = \int d^3z' d^3z'' f_{\hat{n}_1}(z') \left( f_{\hat{n}_2}(z'' - z) - f_{\hat{n}_2}(z'') \right) G_b(z' - z'').
\tag{21}
$$

Transforming to momentum space and using that $G_b(k)$ is even, we have

$$
h_{\hat{n}_1,\hat{n}_2}(z) = \int \frac{d^3k}{(2\pi)^3} \left( \cos(\omega\tau - k_x x - k_y y) - 1 \right) f_{\hat{n}_1}(k) f_{\hat{n}_2}(-k) G_b(k)
\tag{22}
$$

where $f_{\hat{n}}(k)$ is given by

$$
f_{\hat{n}}(k) = \int d^3z e^{i(\omega\tau - k_x x - k_y y)} f_{\hat{n}}(z) = \frac{1}{i\omega - \hat{n}\cdot k}.
\tag{23}
$$

The function $h_{\hat{n}_1,\hat{n}_2}(z)$ can be obtained in closed form for the boson kinetic term of our action. The result is presented in Appendix A. We now have a closed form expression for all fermion $n$-point functions of our theory:

$$\lim_{N_f \to 0} \langle \psi_{i_1}^\dagger(z_1)\psi^{j_1}(w_1)...\psi_{i_n}^\dagger(z_n)\psi^{j_n}(w_n) \rangle = \frac{k_F^{n/2}}{(2\pi)^{3n/2}}$$

$$\times \sum_{\substack{\sigma \in S_n \\ s_1=\pm 1 \\ ... \\ s_n=\pm 1}} \left[ \prod_{l=1}^n \delta_{i_l}^{j_\sigma(l)} \frac{e^{-ik_F s_l |z_{l,\sigma(l)}|+is_l\pi/4}}{\sqrt{|z_{l,\sigma l}|}(-is_l|z_{l,\sigma(l)}|+\tau_{\sigma(l)}-\tau_l)} \right]$$

$$\times H_\lambda(\{s_i \hat{z}_{i,\sigma(i)}\},\{z_i\},\{w_{\sigma(i)}\}) + o(k_F^{n/2}) \qquad (24)$$

where $z_{ij} = w_j - z_i$. Here $o(k_F^{n/2})$ (little-o notation) signify terms subleading to $k_F^{n/2}$ when $k_F$ is large compared to the scale set be the $z_{ij}$. We will use the notation $\langle O \rangle_{k_F^{n/2}}$ to signify expectation values calculated to leading order using this expression.

## 2.3 Density $n$-point functions

The fermion density of species $i$ is given by $\rho_i(z) = \psi_i^\dagger(z)\psi_i(z)$. To use the framework of the previous section to calculate correlation functions of this composite operator it will be necessary to contract $\psi_i^\dagger(z)$ and $\psi_i(z)$ at the same point using the the background field Green's function. We only have the approximate function $G_{\mathrm{IR}}$, which is not valid for length scales of order $1/k_F$ or shorter so it cannot be used for this. We will instead only study correlations of the total fermion density operator that is invariant under the global $U(N_f)$:

$$\rho(z) = \sum_i \psi_i^\dagger(z)\psi_i(z). \qquad (25)$$

In calculating a correlation function $\langle \rho(z_1)\rho(z_2)... \rangle$ using Eq. (4) we sum over the different permutations of the contractions of $\psi_i^\dagger$ and $\psi_i$, and over the flavor indices $i$. The background field Green's function is diagonal in indices so each contraction constrains the sums over flavor indices. One sum over a flavor index will remain for each cycle of the permutation so each permutation $\sigma$ will come with a factor $N_f^{\mathrm{cycles}(\sigma)}$ where cycles$(\sigma)$ is the number of cycles in permutation $\sigma$. In the small $N_f$ limit that we consider we have only kept the leading contribution and we should thus only sum over the permutations with a single cycle since all other permutations are subleading in small $N_f$. For density $n$-point functions with $n > 1$ we will then never contract $\psi_i^\dagger(z)$ and $\psi_i(z)$ at the same point (since that would contribute an extra cycle) unless two of the $z_i$ are equal. The $G_{\mathrm{IR}}$ of the previous section is thus sufficient for calculating correlation functions of $\rho(z)$ to leading order in small $N_f$. See Fig. 2 for an example of this in the case of the density-density correlator. For a fermion density $n$-point function we have:

$$\langle \rho(z_1)...\rho(z_n) \rangle_{k_F^{n/2}} = N_f \frac{k_F^{n/2}}{(2\pi)^{3n/2}}$$

$$\times \sum_{\substack{\sigma \in S_n^{\mathrm{cyclic}} \\ s_1=\pm 1 \\ ... \\ s_n=\pm 1}} \left[ \prod_{i=1}^n \frac{e^{-ik_F s_i |z_{i,\sigma i}|+is_i\pi/4}}{\sqrt{|z_{i,\sigma i}|}(-is_i|z_{i,\sigma i}|+\tau_{\sigma i}-\tau_i)} \right] H(\{s_i \hat{z}_{i,\sigma i}\},\{z_i\},\{z_{\sigma(i)}\}) + \mathcal{O}(N_f^2). \qquad (26)$$

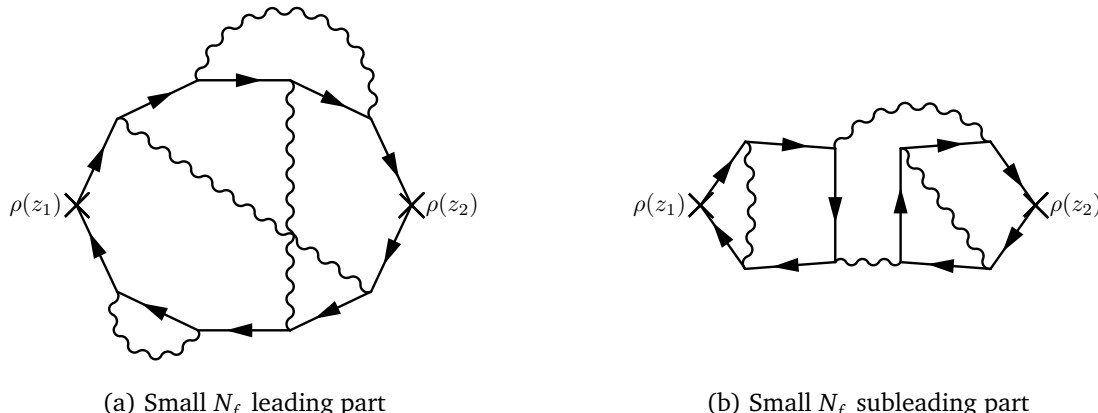

(a) Small $N_f$ leading part

(b) Small $N_f$ subleading part

Figure 2: Once the fermionic fields have been integrated out and the resulting determinant set to 1 (by the small $N_f$ limit) there are two classes of diagrams contributing to the fermion density-density correlator. (a) Shows one of the diagrams in the first class that contributes at order $N_f$ (b) Shows a diagram in the second class that contributes at order $N_f^2$.

## 3 Results

In this section we apply the framework developed in the preceding section to explicitly calculate some observables in the $N_f \to 0$ limit. As a consistency check we expand these results in the coupling constant and compare with perturbation theory in Appendix B.

### 3.1 Fermion two-point function

Since we have rotational symmetry we need only consider a positive separation $r$. We find the real-space fermion two point function:

$$
\begin{aligned}
\langle \psi^\dagger(0)\psi(\tau,r) \rangle_{k_F^{1/2}} = &-\sqrt{\frac{k_F}{r}} \frac{e^{\frac{\lambda^2(\tau^2-r^2)}{12\pi\sqrt{\tau^2+r^2}}}}{2\pi^{3/2}(\tau^2+r^2)} \\
&\times \left[ (\tau-r)\sin\left(k_F r + \frac{\lambda^2 \tau r}{6\pi\sqrt{\tau^2+r^2}}\right) + (\tau+r)\cos\left(k_F r + \frac{\lambda^2 \tau r}{6\pi\sqrt{\tau^2+r^2}}\right) \right].
\end{aligned} \tag{27}
$$

This is equivalent to what is found in [22] and we refer to that work for an in-depth analysis of this fermion two-point function.

### 3.2 Density-density correlator

We note the property of the function $I$:

$$
I_{\hat{n}}(z_1, z_2) + I_{\hat{n}}(z_2, z_3) = I_{\hat{n}}(z_1, z_3). \tag{28}
$$

This, together with the fact that $I_{\hat{n}}(z,z) = 0$ means that there are considerable cancellations in the sum of Eq. (19) for density correlators where some $z_i - z_j$ are parallel to each other for different $i$, $j$. This is true for the density 2-point function and therefore it is given by the rather simple expression:

$$
\langle \rho(0)\rho(\tau,r) \rangle_{k_F^1} = N_f k_F \frac{\tau^2 - r^2 + (\tau^2+r^2)\sin(2k_F r)e^{-\frac{\lambda^2(\tau^2+2r^2)}{3\pi\sqrt{\tau^2+r^2}}}}{4\pi^3 r(\tau^2+r^2)^2}. \tag{29}
$$

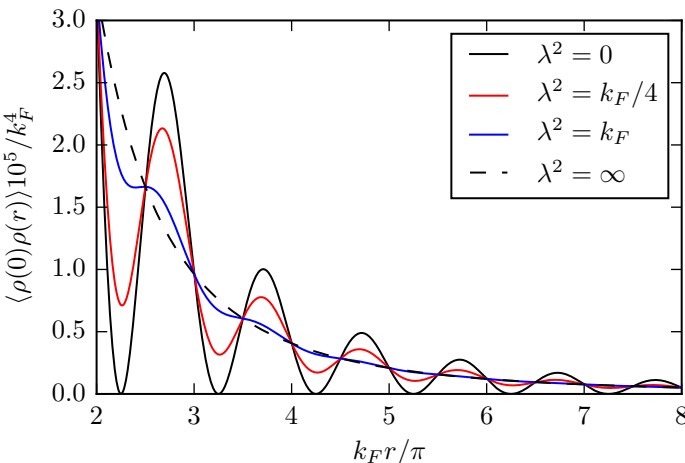

Figure 3: Equal time density-density correlator. This result is only valid for $\lambda^2 \ll k_F$ but note that for any finite $\lambda$ the correlator exponentially approaches the $\lambda = \infty$ case for large separations $r$.

The equal time correlator is given by $|\tau| \ll r$. We can not set $\tau = 0$ directly since this expression is only valid for $\tau \gg k_F^{-1}$, though we see that the limit $|\tau| \ll r$ has the same effect:

$$|\tau| \ll r: \quad \langle \rho(0)\rho(r) \rangle_{k_F^1} = N_f k_F \frac{\sin(2k_F r)e^{-\frac{2\lambda^2 r}{3\pi}} - 1}{4\pi^3 r^3}. \tag{30}$$

For $\lambda = 0$ we see the familiar Friedel oscillations with wave-vector $2k_F$ and a power-law decay. For a finite coupling $\lambda$ the oscillations decay exponentially in the separation $r$ with decay length set by $1/\lambda^2$. See Fig. 3. For separations longer than $1/\lambda^2$ we have

$$\langle \rho(0)\rho(\tau,r) \rangle_{k_F^1} \approx \langle \rho(0)\rho(\tau,r) \rangle_{\text{IR}} \equiv N_f k_F \frac{\tau^2 - r^2}{4\pi^3 r (\tau^2 + r^2)^2}. \tag{31}$$

In the large separation limit the scale $\lambda^2$ drops out and the IR behavior of the density-density correlator is independent of $\lambda^2$. The value of $\lambda^2$ only sets the scale of a crossover to this IR behavior. In momentum space this IR correlator is

$$\langle \rho(k_1)\rho(k_2) \rangle_{\text{IR}} = \delta^3(k_1 + k_2) \left( -N_f k_F \frac{|\omega_1|}{2\pi \sqrt{\omega_1^2 + k_{x,1}^2 + k_{y,1}^2}} + C \right). \tag{32}$$

This is simply the one-loop self-energy correction to the boson, in the limit of small energies *and* momenta. The Fourier transforms in the spatial and temporal directions do not commute, different orderings differ by the undetermined constant $C$. The same ambiguity arises when calculating the self-energy correction perturbatively, there manifesting itself as energy and momentum integrals not commuting. The interpretation of this is discussed in more details in [8].

The result in (29) can be understood by considering the infinite sum of diagrams contributing to it. Each diagram will have two fermion lines, one going from the insertion of $\rho(0)$ to $\rho(\tau,r)$ and one in the opposite direction. A general diagram will have many boson exchanges along these lines. Let us consider the diagrams in momentum space. Each fermion propagator will have its momentum close to the Fermi surface for low-energy processes. The boson will carry a momentum much smaller than $k_F$ for the dominant processes. Therefore

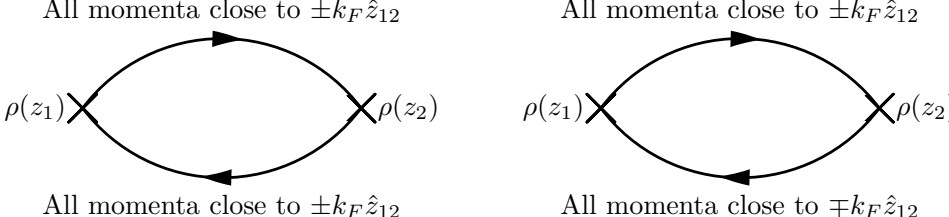

Figure 4: These two diagrams correspond to the two dominant classes of momentum configurations for large separations $z_2 - z_1$. Here we imagine an infinite series of boson exchanges attached in all possible combinations on the upper and lower lines. Since the boson carries momentum much smaller than $k_F$, this will still keep the fermion momenta in the same patch along the upper and lower lines. The two lines however need not belong to the same patch. Since only two opposite patches dominate in the large separation limit the upper and lower lines are either in the same (a) or opposite patches (b). In the former case the dominant external momentum is small compared to $k_F$, in the latter case the dominant external momentum is close to $\pm 2k_F \hat{z}_{12}$.

each of these two fermions lines will have their momenta confined to one patch each of the Fermi surface. We know from section 2.1 that momenta in patches in the directions parallel to $z_{12} = z_2 - z_1$ will give the dominant contribution when we go back to real space. There are thus four dominant regions of the multidimensional momentum space associated to each diagram. All momenta on the line from $\rho(z_1)$ to $\rho(z_2)$ can be either in the patch close to $-k_F \hat{z}_{12}$ or $k_F \hat{z}_{12}$, and similarly for the momenta on the line from $\rho(z_2)$ to $\rho(z_1)$. See Fig. 4. We separate the contributions from processes where the two patches are the same, $G_{\rho\rho}^+$, and where they are opposite, $G_{\rho\rho}^-$:

$$\langle \rho(0)\rho(\tau,r)\rangle_{k_F^1} = G_{\rho\rho,k_F^1}^+(\tau,r) + G_{\rho\rho,k_F^1}^-(\tau,r). \tag{33}$$

Processes with opposite patches have external momenta $k \approx \pm 2k_F \hat{z}_{12}$ and are thus the oscillating part of Eq. (29) while processes where the patches are the same have external momenta $k \ll k_F$:

$$G_{\rho\rho,k_F^1}^+(\tau,r) = N_f k_F \frac{\tau^2 - r^2}{4\pi^3 r (\tau^2 + r^2)^2} \tag{34}$$

$$G_{\rho\rho,k_F^1}^-(\tau,r) = N_f k_F \sin(2k_F r) \frac{e^{-\frac{\lambda^2(\tau^2 + 2r^2)}{3\pi\sqrt{\tau^2 + r^2}}}}{4\pi^3 r (\tau^2 + r^2)}. \tag{35}$$

The non-oscillating part, $G_{\rho\rho,k_F^1}^+(\tau,r)$, receives no corrections from interactions at all. This is expected since the diagrams contributing to this are completely symmetrized fermionic loops. It was shown by Feldman et. al that the leading contribution to this, as $\omega_i, k_i \ll k_F$, cancels out completely in the symmetrized sum (this point of their calculation was specifically pointed out in [25]). Only the non-interacting diagram is not a symmetrized fermionic loop and it survives the cancellation.

The oscillating part, $G_{\rho\rho,k_F^1}^-(\tau,r)$, does not have this cancellation since now two of the vertices in the fermionic loop have momenta of order $k_F$. Here, however, it turns out that the sum of all these diagrams exponentiates and for large separations they completely cancel. The density-density large separation result above is therefore exactly what is obtained by a simple one loop calculation, only taking into account the process where both fermions are on

the same patch. The fact that all processes with opposite patches cancel out is, however, non-trivial and requires the above non-perturbative calculation to be seen. The exponentiation and subsequent cancellation for large separations is the main new result obtained from applying the framework of the previous section to the density-density correlator.

## 4 Conclusion

The limit of low energies compared to the Fermi energy constrains fermions to live very close to the Fermi surface and only scatter in the forward direction. This makes the fermions almost one-dimensional. This "hidden" one-dimensionality and its consequences have been noted before, see [26] for an overview. We write *almost* one-dimensional because for some processes the fermions still see the curvature of the Fermi surface. There are, however, sectors where the curvature is not seen and the fermions can be exactly described as one-dimensional, albeit coupled to a two-dimensional boson. The $N_f \to 0$ limit singles out this sector precisely. Only the fermionic loops see the curvature. Studying the effectively one-dimensional fermions in the $N_f \to 0$ limit allowed us to find a closed form expression for the fermion $n$-point functions. In Section 3.2 we used this expression to calculate the fermion density two-point function.

The physics of the $N_f \to 0$ limit quantum critical metal is not expected to be similar to the, currently intractable, finite $N_f$ case. We can however use it to gain some insights into how the different diagrams of the full perturbation theory behave as we saw for the density-density correlator. It provides a contrasting alternative to the studies in the opposite limit of large $N_f$ [10–13] and it provides an efficient way of calculating high order diagrams that are also part of the perturbative expansion of the finite $N_f$ case.

The, perhaps surprising, exponential decay of Friedel oscillations is distinct from the power-laws found in Fermi liquids. The density-density correlator has been studied before in strongly coupled systems using some of the approaches mentioned in the introduction [19,27,28]. The exponential damping has not been found in these works, however, [27–29] find that interaction effects suppress the $2k_F$ oscillations.

A phenomenon similar to the exponential decay seen here has been found in holographic models of strongly interacting fermions in 2+1 dimensions. It is not entirely clear how to obtain a holographic state of a strongly interacting Fermi surface. Several different approaches have been used. Probe fermions [30,31] are very similar to the $N_f \to 0$ limit studied here in that there is no back-reaction from the fermion but the fermion still receives non-perturbative corrections from gapless excitations. However, this means that the presence of the fermion is not seen in holographic current correlators. $2k_F$ singularities are also not seen in electron star geometries, where fermion back-reaction is taken into account [32]. A more recent paper [33] studied density correlators in the Reissner-Nordström dual for complex momenta and found a branch-cut terminating at a complex momentum at zero-temperature. This gives rise to ex-ponentially damped oscillations of the density-density correlator. In [34] the authors consider the susceptibility of both the Reissner-Nordström black brane and also that of a "3-charge black brane". They find damped oscillations in both cases and they further compare the period with the Fermi momentum found in the fermion spectral function of these models [35] and find that the oscillations do not occur at $2k_F$, but closer to $1k_F$. It is therefore not entirely clear that these oscillations are related to a Fermi surface and it is certainly not certain that the ex-ponential damping they see is the same phenomenon as what is found in this paper. However, it is an interesting prospect so let us for a moment consider these two observations to be re-lated. The damping rates of the holographic models are of the order of the Fermi momentum, $l_d^{-1} = \mathrm{Im}(k^*) \sim k_F$, whereas in our model we have $l_d^{-1} \sim \lambda^2$. We consider a dimensionful coupling $\lambda^2 \ll k_F$ whereas the coupling in [33,34] is dimensionless and is taken to infinity,

likely more similar to the $k_F \ll \lambda^2$ case. If we are indeed seeing the same phenomenon then we would expect the damping rate of our model, $l_{\mathrm{d}}^{-1}(\lambda^2)$, to go as $\lambda^2$ for $\lambda^2 \ll k_F$ but then saturate to $\sim k_F$ once $\lambda^2 \sim k_F$ where our theory breaks down.

Whether this $T = 0$ exponential damping is a general feature of strongly interacting fermions at finite density or a consequence of the $N_f \to 0$ limit and the specifics of holographic theories is too early to tell but it is an interesting question to explore in the future.

Several instabilities occur in Fermi liquids so it would not be surprising if this theory also shows one of them. An important future direction of research is to search for instabilities of this model, and generalizations of it, using the framework developed here.

# Acknowledgements

The author wishes to thank Andrey Bagrov, Alexander Balatsky, Bartosz Benenowski, Blaise Goutéraux, Oscar Henriksson, Nick Poovuttikul, Koenraad Schalm and Konstantin Zarembo for useful discussions. The author would additionally like to thank Koenraad Schalm for reading and giving detailed comments on an early version of this manuscript. This work was supported in part by a VICI (Koenraad Schalm) award of the Netherlands Organization for Scientific Research (NWO), by the Netherlands Organization for Scientific Research/Ministry of Science and Education (NWO/OCW), by the Foundation for Research into Fundamental Matter (FOM), by Knut and Alice Wallenberg Foundation and by Villum Foundation.

# A   Calculating $h_{\hat{n}_1,\hat{n}_2}(z)$

In this section we calculate the function $h_{\hat{n}_1,\hat{n}_2}(z)$ as defined in Eq. 22. Up till now we have not had to consider the form of the boson propagator. To continue we need to use the specific form of the free boson propagator of our theory:

$$G_b(k) = \frac{1}{\omega^2 + k_x^2 + k_y^2}. \tag{36}$$

We need to perform three integrals to find $h$. First we make the change of variables $k = r\hat{n}_1 \times \hat{n}_2 + rs_2 z \times \hat{n}_1 + rs_3 z \times \hat{n}_2$. After integrating $r$ over all of $\mathbb{R}$ we have

$$
\begin{aligned}
h_{\hat{n}_1,\hat{n}_2}(z) = -\int & \frac{\mathbb{S}_2\mathbb{S}_3}{(2\pi)^3} \frac{\pi|\tau \hat{n}_1 \times \hat{n}_2 \cdot z|^3}{(n_s \times z - \hat{n}_1 \times \hat{n}_2)^2} \\
& \times \frac{1}{((1 + is_2\tau)\hat{n}_1 \times \hat{n}_2 + \tau n_s \times \hat{\tau}) \cdot z} \cdot \frac{1}{((1 - is_3\tau)\hat{n}_1 \times \hat{n}_2 + \tau n_s \times \hat{\tau}) \cdot z}
\end{aligned}
\tag{37}
$$

where $n_s = s_2\hat{n}_1 + s_3\hat{n}_2$ and $\hat{\tau} = (1,0,0)$. We can now do the $s_2$ integral using the residue theorem. The denominator is a fourth order polynomial in $s_2$. Two roots are polynomials in $s_3$ whereas the the remaining second order polynomial in $s_2$ has roots in terms of radicals of $s_3$. The contribution from the first two poles can thus easily be integrated once again, now over $s_3$, since it is a rational function. The range is no longer $\mathbb{R}$ since the pole in $s_2$ will leave the upper half plane where we close the $s_2$-contour for certain values of $s_3$. The contribution from the last two poles is more involved because of the radicals. One of these poles is always in the UHP and the other in the LHP so we only need to account for one of them. By making the change of variables

$$s_3 \mapsto \frac{\sqrt{\tau^2 + y^2}\sinh(w) - x}{\tau^2 + x^2 + y^2}, \tag{38}$$

we get rid of the radicals and can carry out the $w$ integral. In the end the total result can be written as

$$
\begin{aligned}
h_{\hat{n}_1,\hat{n}_2}(z) = \frac{1}{4\pi(1 - \hat{n}_1 \cdot \hat{n}_2)} & \bigg[ |\hat{n}_1 \diamond z| + |\hat{n}_2 \diamond z| - 2r \\
& - 2\frac{\tau \hat{n}_1 \cdot \hat{n}_2 - i(\hat{n}_1 \cdot z + \hat{n}_2 \cdot z + i\tau)}{|\hat{n}_1 \diamond \hat{n}_2|} \\
& \times \Big(\pi\theta(-\tau) + i\,\mathrm{sgn}(\hat{n}_l \diamond z)\log(A) + i\big[\theta(-\hat{n}_1 \diamond z) - \theta(\hat{n}_2 \diamond z)\big] \\
& \quad \times \log\Big[\frac{i\tau\hat{n}_1 \diamond \hat{n}_2 + \hat{n}_1 \diamond z - \hat{n}_2 \diamond z}{\hat{n}_1 \diamond \hat{n}_2(\hat{n}_k \cdot z + i\tau)}\Big]\Big)\bigg],
\end{aligned}
\tag{39}
$$

where

$$z = (\tau, x, y) \tag{40}$$

$$n_i = (x_i, y_i) \tag{41}$$

$$r = \sqrt{\tau^2 + x^2 + y^2} \tag{42}$$

$$\tilde{r} = \text{sgn}(\hat{n}_l \diamond z) r \tag{43}$$

$$\hat{n}_1 \cdot \hat{n}_2 = x_1 x_2 + y_1 y_2 \tag{44}$$

$$\hat{n}_1 \diamond \hat{n}_2 = x_1 y_2 - y_1 x_2 \tag{45}$$

$$\hat{n}_1 \diamond z = x_1 y - y_1 x \tag{46}$$

$$k = \begin{cases} 1, & \text{for} \quad \hat{n}_1 \diamond \hat{n}_2 < 0 \\ 2, & \text{for} \quad 0 < \hat{n}_1 \diamond \hat{n}_2 \end{cases} \tag{47}$$

$$l = 3 - k \tag{48}$$

and

$$A = \frac{(\hat{n}_1 \cdot \hat{n}_2 - 1)(\tau(1 + \hat{n}_1 \cdot \hat{n}_2) - i(\hat{n}_1 + \hat{n}_2) \cdot z)}{(\hat{n}_1 \diamond \hat{n}_2)^2 (i\tau + \hat{n}_1 \cdot z)(i\tau + \hat{n}_2 \cdot z)}$$
$$\times \left( i\tilde{r}\tau |\hat{n}_1 \diamond \hat{n}_2| + (\hat{n}_l \diamond z - \tilde{r})(\hat{n}_k \diamond z + \tilde{r}) + \tau^2 \hat{n}_1 \cdot \hat{n}_2 - i\tau(\hat{n}_1 \cdot z + \hat{n}_2 \cdot z) + \tau^2 \right)^{1/2}. \tag{49}$$

We see that the prefactor of this expression diverges for $\hat{n}_1 = \hat{n}_2$ and this case has to be treated separately. The function is continuous at this point however, and we can simply take the limit $\hat{n}_1 = \hat{n}_2$ to obtain

$$h_{\hat{n}, \hat{n}}(z) = \frac{r^3 - |\hat{n} \diamond z| \left( 3(\hat{n} \cdot z)^2 + 3i\tau \hat{n} \cdot z + (\hat{m} \cdot z)^2 \right)}{12\pi(\hat{n} \cdot z + i\tau)^2}. \tag{50}$$

This can be compared to the calculation for the two-point function. There we have $\hat{n} \diamond z = 0$, and using this we get

$$h_{\hat{n}, \hat{n}}(z) = \frac{r^3}{12\pi(\hat{n} \cdot z + i\tau)^2} = \frac{(\hat{n} \cdot z - i\tau)^2}{12\pi\sqrt{(\hat{n} \cdot z)^2 + \tau^2}}. \tag{51}$$

This agrees with the previous result of [22]. For the density-density correlator we additionally need $h$ for $\hat{n}_1 = -\hat{n}_2$ and $n_1 \diamond z = 0$. Taking the limits simultaneously we find

$$h_{\hat{n}, -\hat{n}}(z) = -\frac{\sqrt{\tau^2 + (\hat{n} \cdot z)^2}}{4\pi}. \tag{52}$$

We note that $A$ diverges as $\sim 1/\hat{n}_1 \diamond \hat{n}_2$ for $\hat{n}_1 = -\hat{n}_2$. The prefactor of the logarithm is proportional to $\hat{n}_1 \diamond z$:

$$h_{\hat{n}_1, \hat{n}_2}(z) = \frac{|\hat{n}_1 \diamond z| \log(|\hat{n}_1 \diamond \hat{n}_2|)}{4\pi} + \text{finite}. \tag{53}$$

We have an additional constraint in Eq. (20) however, the $n_i$ are parallel or anti-parallel to $w_i - z_i$. Using this one can show that this divergence in $h$ as $\hat{n}_1$ and $\hat{n}_2$ become anti-parallel cancels out in the sum of Eq. (20) and is not seen in observables.

## B  Perturbative verification

To verify our non-perturbative results we can expand in the coupling constant $\lambda$ and compare with perturbation theory. We do so here for both the fermion two-point function and the density-density correlator, both up to order $\lambda^2$. Since our non-perturbative results are only valid at long wavelengths and low energies, we will expand around singularities in momentum space and verify that the leading singularities agree with perturbation theory. We start off by verifying the two-point function (27) at tree level:

$$\langle\psi^\dagger(0)\psi(\tau,r)\rangle_{k_F^{1/2},\lambda^0} = -\frac{\sqrt{\frac{k_F}{r}}((\tau-r)\sin(k_F r)+(\tau+r)\cos(k_F r))}{2\pi^{3/2}(\tau^2+r^2)}. \tag{54}$$

In momentum space this is (here and henceforth we omit the momentum-conserving $\delta$-function)

$$\langle\psi^\dagger\psi(\omega,k)\rangle k_F^{1/2},\lambda^0 = -\sqrt{\pi k_F}(1+i\,\mathrm{sgn}(\omega))\int_0^\infty dr\sqrt{r}J_0(kr)e^{-r|\omega|+ik_F r\,\mathrm{sgn}(\omega)} \tag{55}$$

where $J_0$ is the zeroth Bessel function of the first kind. We look for singularities and therefore want the integral to diverge. The only possibility for this is when $\omega = 0$, so there is no exponential decay, and when $k = k_F$, so the oscillations of the Bessel function cancels those of the exponential. To expand around this point we can approximate the Bessel function with its asymptotic oscillatory behaviour. In doing so we only modify the finite part of the integral. We find

$$\langle\psi^\dagger\psi(\omega,k_F+k_x)\rangle_{k_F^{1/2},\lambda^0} = \left(1-\frac{i\omega}{2k_F}\right)\frac{1}{i\omega-k_x}+\frac{1}{8k_F}\log\left(\frac{k_F}{i\omega-k_x}\right)+\text{finite}. \tag{56}$$

The full fermion two-point function of our toy model at tree level is given by

$$\langle\psi^\dagger\psi(\omega,k)\rangle_{\lambda^0} = \frac{1}{i\omega-k^2/(2k_F)+k_F/2}$$

$$\approx \frac{1}{i\omega-(k-k_F)} \equiv G_0^{\text{patch}}(\omega,k-k_F) \tag{57}$$

We see that the results agree to leading order for $\omega \ll k_F$ and $k$ close to $k_F$. This same technique is used below to find the leading divergences of the density-density correlators and also of the higher order in $\lambda$ contributions to the correlators. The two-point function, Eq. (27), expanded to second order in $\lambda$ is found to be

$$\langle\psi^\dagger\psi(\omega,k_F+k_x)\rangle_{k_F^{1/2},\lambda^2} = -\lambda^2\frac{\sqrt{k_x^2+\omega^2}}{4\pi(k_x-i\omega)^3}+\text{subleading}. \tag{58}$$

This is to be compared to the diagram in Fig. 5. Evaluating this diagram with all fermion propagators linearized at the patch of the Fermi surface at $k = (k_F+k_x)\hat{x}$ gives

$$D_5 = \lambda^2 G_0^{\hat{x}}(\omega,k_x)^2\int\frac{d\omega_1 dq_x dq_y}{(2\pi)^3}G_0^{\hat{x}}(\omega+\omega_1,k_x+q_x)G_b(\omega_1,q)$$

$$= \lambda^2\frac{\sqrt{k_x^2+\omega^2}}{4\pi(i\omega-k_x)^3} \tag{59}$$

where $G_0^{\hat{x}}$ is defined in Eq. 57. The density-density correlator (29) at order $\lambda^0$ is easily verified in real space using (27) at the same order. To verify the density-density correlator at order $\lambda^2$ we need to calculate the three diagrams in Fig. 6. Diagram 6ab can similarly be obtained by

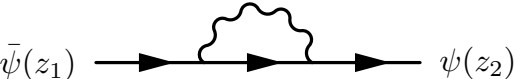

Figure 5: The single diagrams that contributes at order $\lambda^2$ to the fermion two-point function.

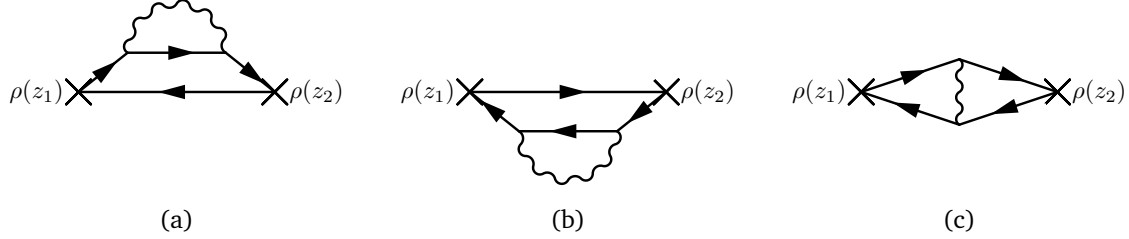

Figure 6: The three diagrams that contribute at order $\lambda^2$ to the density-density correlation function.

combining the fermion two-point function at order $\lambda^0$ and $\lambda^2$ in real space:

$$D_{6a}(\tau, r) = -N_f G_{\lambda^0}(\tau, r) G_{\lambda^2}(-\tau, r), \tag{60}$$

$$D_{6b}(\tau, r) = -N_f G_{\lambda^2}(\tau, r) G_{\lambda^0}(-\tau, r). \tag{61}$$

Subtracting these two diagrams in real space from Eq. (29) and Fourier transforming we find for small $\omega$ and $k \approx 2k_F$:

$$\langle \rho\rho(\omega, k)\rangle_{k_F, \lambda^2, N_f^1} - D_{6ab}(\omega, k) = -\frac{N_f \lambda^2 \sqrt{k_F}}{4\pi^3 \sqrt{|\omega|}} \text{Re}\left[(1+i)K\left(\frac{i(|k|-2k_F)+|\omega|}{2|\omega|}\right)\right]$$
$$+ \text{subleading.} \tag{62}$$

where $K$ is the complete Elliptic integral of the first kind. Calculating Diagram 6c requires a more involved calculation. Since we are interested in the external momentum $k = (2k_F + k_x)\hat{x}$, $|k_x| \ll k_F$, we can expand the fermion propagators in momenta in the patches at $\pm k_F \hat{x}$. After this we can perform all the momentum integrals to obtain:

$$D_{6c} = -N_f \lambda^2 \int \frac{d^3 k_1}{(2\pi)^3} \frac{d^3 q}{(2\pi)^3} G_0^{\hat{x}}((2k_F + k_x)\hat{x} + k_1)$$
$$\times G_0^{\hat{x}}((2k_F + k_x)\hat{x} + k_1 + q)G_0^{-\hat{x}}(k_1 + q)G_0^{-\hat{x}}(k_1)G_b(q)$$
$$= N_f \lambda^2 \sqrt{k_F} \int \frac{d\omega_B d^2 q}{(2\pi)^3} \frac{\sqrt{k_x + i\omega} - \sqrt{k_x + q_x - i(\omega_B - \omega)}}{\pi\left(q_x^2 + \omega_B^2\right)} G_b(\omega_B, q) + \text{subleading}$$
$$= -\frac{N_f \lambda^2 \sqrt{k_F}}{4\pi^3 \sqrt{|\omega|}} \text{Re}\left[(1+i)K\left(\frac{ik_x + |\omega|}{2|\omega|}\right)\right] + \text{subleading.} \tag{63}$$

We have thus verified the density-density correlator at order $\lambda^2$.

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
