# Peer review of "A Framework for Studying a Quantum Critical Metal in the Limit $N_f\rightarrow0$"

_SciPost Physics, doi:SciPost Phys. 4, 015 (2018)_

## Round 1 · Referee Report · Anonymous · 2017-12-20

Strengths
1. Treatment of the problem is technically sound.
2. Demonstration of a non-perturbative effect of the fermion-boson interaction in the density-density response.
3. The introduction contains a good exposition of the problem.
Weaknesses
1. The conclusion foregoes a discussion of the principal result with respect to the physical properties of the system at hand.
Report
The author studies the correlation functions of a two-dimensional metal which is strongly coupled to gapless bosons. Building on earlier related work, the present publication elaborates on the density-density response in a strictly controllable, albeit somehow arbitrary limit which is non-perturbative in the coupling strength. As the main result, the density-density correlation features an exponentially fast extinction in real space, which appears only beyond perturbation theory.
I can recommend the publication, but I suggest to elaborate in better detail on the significance of the result for critical metals in two dimensions.
Requested changes
The author might consider to widen the scope of the discussion to include some implications of their fast decay. What does this mean for the stability of the critical theory at Q=0, and for the stability of approximations where the Landau damping is incorporated from the outset? Can the improved density-density correlator be used to supplement previous N_f->0 methods?
Author: Petter Säterskog on 2018-01-31 [id 209]
(in reply to Report 1 on 2017-12-20)
Dear referee, Thank you for your comments.
1: "What does this mean for the stability of the critical theory at Q=0, and for the stability of approximations where the Landau damping is incorporated from the outset?" In order to thoroughly study instabilities of the $Q=0$ quantum critical metals I have found it necessary to do several extensions to this model and some rather complicated calculations so I have decided to do that in a follow up work and limit this paper to introducing the framework for calculating $N_f\rightarrow0$ correlation functions. I have now added a sentence at the end of the conclusion to underline the importance of searching for instabilities.
2: "Can the improved density-density correlator be used to supplement previous N_f->0 methods?" The density-density correlator indeed gives part of the $N_f$ corrections to the boson. However, the relevant boson self-energy contribution comes from small momenta and as we argue at the top of p. 13 (see ref. 25), all corrections beyond one loop cancel out so we can not use this result to improve on the one loop boson self-energy. The new non-perturbative effects found in the density-density correlator all appear at large momentum $2k_F$ and are thus not useful for improving on the boson self-energy.
Best regards, Petter
Anonymous on 2020-10-08 [id 998]
In this context, I would like to bring to your attention my papers: https://arxiv.org/abs/1608.01320 (Superconducting instability in non-Fermi liquids) and https://arxiv.org/abs/1608.06642.
Please ignore this message if you already knew about these works, but did not find them relevant to include in the references.

---

## Round 2 · Referee Report · Anonymous (Referee 1) · 2018-2-15

Report

The author's response addressed the main points of criticism regarding the previous draft, clarifying the issues at hand. I can recommend publication in the present form.

---

## Round 2 · List of Changes

Added comment about importance of searching for instabilities.
Added missing definition of $r$ for Eq. 39.

---

## Editorial Decision

published